# PRIME-3D2D is a 3D2D model to predict binding sites of protein–RNA interaction

Juan Xie[1,2], Jinfang Zheng[1,2], Xu Hong[1], Xiaoxue Tong[1] & Shiyong Liu [1✉]

Protein-RNA interaction participates in many biological processes. So, studying protein–RNA interaction can help us to understand the function of protein and RNA. Although the protein–RNA 3D3D model, like PRIME, was useful in building 3D structural complexes, it can't be used genome-wide, due to lacking RNA 3D structures. To take full advantage of RNA secondary structures revealed from high-throughput sequencing, we present PRIME-3D2D to predict binding sites of protein–RNA interaction. PRIME-3D2D is almost as good as PRIME at modeling protein–RNA complexes. PRIME-3D2D can be used to predict binding sites on PDB data (MCC = 0.75/0.70 for binding sites in protein/RNA) and transcription-wide (MCC = 0.285 for binding sites in RNA). Testing on PDB and yeast transcription-wide data show that PRIME-3D2D performs better than other binding sites predictor. So, PRIME-3D2D can be used to predict the binding sites both on PDB and genome-wide, and it's freely available.

---

[1] School of Physics, Huazhong University of Science and Technology, Wuhan, Hubei 430074, China. [2]These authors contributed equally: Juan Xie and Jinfang Zheng. ✉email: liushiyong@gmail.com

In recent years, many noncoding RNAs[1] were discovered by next-generation sequencing (NGS), without knowing the function of these noncoding RNAs. RNA never acts alone, and it works with RNA-binding proteins (RBPs) or other molecules[2]. Protein–RNA interaction participates in several cellular processes like splicing, mRNA location, gene regulation[3,4]. In previous studies, a lot of RBPs were discovered without interaction partner information[5]. Studies on RNA secondary structurome uncover many RNA secondary structures in vivo or in vitro[6,7] in which the conclusions imply that RNA secondary structure plays a significant role in protein–RNA interaction.

Many studies have been carried out to investigate the protein–RNA interaction. High-throughput experimental techniques, such as HITS-CLIP (High Throughput Sequencing of RNA isolated by Crosslinking Immunoprecipitation)[8], PAR-CLIP (Photoactivatable-Ribonucleoside-Enhanced Crosslinking and Immunoprecipitation)[9], iCLIP (individual-nucleotide resolution UV-Crosslinking and Immunoprecipitation)[10], and eCLIP (enhanced Crosslinking and Immunoprecipitation)[11], provide the protein–RNA interaction data on genome wide. These data have been collected by several databases for further analysis[12–15]. Besides, some computational methods have been proposed for predicting the protein–RNA interaction pairs. The sequence features (such as physicochemical properties of protein and RNA, sequence composition features, motif information), RNA secondary information or RNA 3D structure features were applied to these methods[16–19] to predict the protein–RNA interaction. Different to these methods, some other methods were proposed based on networks[20–23] to predict the interaction. In addition, the co-evolution methods were also introduced to predict the 3D protein–RNA complexes[24] or interaction[25]. All these approaches do not take into account the RNA structurome data produced by NGS. However, NGS data can extend our knowledge of protein–RNA interaction on genome wide. Furthermore, we still lack the researches based on 3D information on genome wide because of the lack of a protein–RNA complexes[26,27].

Obtaining the binding sites of protein–RNA interaction is very helpful in understanding its biological functional mechanism. Although high-throughput sequencing methods can obtain a large amounts of RNA-binding sites for specific proteins[28–30], it is still difficult to obtain RNA-binding sites information for all proteins. Therefore, predicting binding sites by computational methods can compensate for this defect. Recently, many teams are working to predict protein–RNA-binding sites. Current methods for predicting protein–RNA interaction binding sites are divided into two major categories: predicting the RNA-binding sites on proteins[31–45], and predicting the protein-binding sites on RNA[46–51]. These methods usually consider the sequence, structure or physicochemical characteristics of the given protein or RNA. The methods for predicting the binding sites of RNA on proteins are mainly divided into two categories: sequence-based and structure-based. The sequence-based approaches mainly use the sequence features of proteins and machine learning methods to identify RNA-binding residues on proteins. For example, BindN utilized the pKa value of side chain, hydrophobicity index and molecular mass of an amino acid as features[33]. Subsequently, Naive Bayes and the identity information of amino acid sequences were employed to predict binding sites on protein[32]. Other groups used PSSM evolutionary information to predict RNA-binding sites on proteins[31,35,39]. Subsequently, in 2010, several teams developed methods to predict RNA-binding sites on proteins. Such as Ma et al.[37] predicted the binding sites of RNA on proteins by using the information of predicted secondary structures, the polarity-charge correlation physicochemical properties and the hydrophilic and hydrophobic properties of amino acids. NAPS used protein sequence characteristics to predict RNA/DNA-binding sites on protein[38]. PiRaNhA used the protein sequence with the position-specific scoring matrices, residue interface propensity, predicted residue accessibility and residue hydrophobicity characteristics to predict the binding sites on RNA-binding residues[34]. Meta-predictor was based on the features used in the other three prediction methods[44] and RNABindR2.0 was based on sequence homology and SVM classifier[36]. Earlier structure-based methods used multiple scores to predict the RNA-binding sites on proteins[40,42]. Some research groups utilized protein structural features to predict RNA-binding sites[52,53]. The template-based approaches predicting the RNA-binding sites on protein were compared the target protein structure with the known complex structure[41,45]. Ren's method was different from previous approaches. The surrounding patches were compared with template patches and the accumulated distances was used as structural features[43].

With the development of high-throughput sequencing methods, thousands of protein-binding sites on RNA have been discovered. Protein-binding site predictors based on sequencing data were also developed[46–51]. However, these methods currently train the model mainly for specific proteins, so they are not universal. In addition, the current binding site prediction tools can only predict the binding sites on the protein or RNA. Therefore, a universal model to predict protein–RNA-binding sites on both protein and RNA is needed.

In this study, PRIME-3D2D is introduced to expand the available structural data of protein–RNA interaction, for overcoming the lack of protein–RNA 3D structure. The protein–RNA 3D3D model is transformed into 3D2D model (the RNA 3D structure is replaced by RNA 2D structure). 3D2D score is introduced in searching templates to describe the binding mode of two complexes. First, the phase transition points are determined in all-to-all pairwise alignment for identifying the good template. Then, PRIME-3D2D based on TM-align[54] and LocARNA[55] is introduced to model the 3D structure and predict the binding sites of protein–RNA interaction. For binding model predictions, benchmarked in 439 binary complexes (NRBC439[27]), the success rate of PRIME-3D2D is almost as good as PRIME for top 10 predictions[27]. For binding sites predictions, benchmarked in NRBC439, PRIME-3D2D obtains the MCC about 0.70. Comparing to the state-of-the-art methods, PRIME-3D2D outperforms other binding-sites predictors on both PDB and genome wide data.

## Results
**Principle of RNA2dA and comparison with LocARNA.** For investigating the effect of RNA secondary structure in RNA alignment, we developed RNA2dA, an RNA alignment approach combined RNA secondary structure and sequence. For RNA, a novel representation called BEAR encoding RNA secondary structure was chosen to represent the RNA. In this paper[56], the software BEAR converts the dot-bracket notation to BEAR encoding. In order to align the RNAs based on this novel representation, a BLOSUM like matrix is calculated from Rfam. In Supplementary Fig. 1, it shows the hot map of RNA-BLOSUM80. The weight combined score matrix of RNA sequence (NUC.4.4) and score matrix of RNA secondary structure (RNABLOSUM80) is determined as 0.2 since the result of benchmark is the best (Supplementary Fig. 2). The result indicates a better RNA aligner should consider both the RNA sequence and secondary structure. In Supplementary Fig. 3, it shows the distribution of SPS when RNA2dA and locARNA were benchmarked in BraliBase II[57]. The mean SPS of RNA2dA is 0.94, which is almost as good as LocARNA. Besides, RNA2dA also achieves a comparable result with Beagle[58].

**Comparing alignment approaches in searching templates**. The previous study on RNA alignment indicated that approach using secondary structure alone performed worse than approach combining RNA sequence and secondary structure information in searching templates[56]. In this section, we can make a conclusion that more templates can be found by the alignment method combining sequence with secondary structure than RNA secondary structure information alone. We performed NRBC90 vs NRBC349 pairwise comparison of protein–RNA binary complexes in NRBC439 set. The similarity of binding mode is measured by interaction RMSD (iRMSD[59]). TM-align was employed to implement protein structure alignment, and LocARNA and RNA2dA (see Methods) were utilized for RNA alignment, which are corresponded to the approaches combining RNA sequence with secondary structure information and using secondary structure alone (bonus is set to 0 in RNA2dA).

In Supplementary Fig. 4, it shows the result of comparison in searching templates for RNA sequence/secondary structure alignment and secondary structure alignment. The curve labeled "locarna-template" is always above the curve labeled "RNA2dA-template", suggesting that RNA alignment approach combining RNA sequence/secondary structure can detect more templates than RNA secondary structure alone. This result can be explained by RNA sequence sometime effects in protein–RNA interaction. For this result, RNA alignment tool LocARNA will be used as template searching algorithm to identity templates in the following study.

**The similarity of binding mode vs that of monomer structures**. The small similarity value of two monomers (RNA and protein) was employed to measure the similarity of binary complexes on the previous studies[27,60]. This may be not the best way to describe the similarity of binary complexes, because we found that the success rate was not the highest (missing some correct models) after applying a cutoff in modeling[27]. In the current study, a scoring function, which combined the protein similarity and RNA secondary structure similarity (3D2D score), was used to measure the similarity of binary complexes. We performed all-to-all paired comparison in NRBC439 set and found that binding mode similarity correlates with the similarity of the participating protein and RNA, with different weights and different phase transition values. We found that the trend of the protein–RNA structural similarity with the binding mode is similar to that of the protein–RNA complexes[27] and protein–protein complexes[60].

The relationship between binding mode and the similarity of monomers are plotted in Fig. 1, with different $W$ values. The subfigure labeled "$W = 1$" is correspond to the similarity of binary complex described by TM-score. The subfigure labeled "$W = 0$" stands for the similarity of binary complex described by the RNA secondary structural identity alone. Other $W$ values correspond to a combination of TM-score and RNA secondary structural identity. When $W$ varies from 1 to 0, the noise signals (points above the iRMSD $\geq 10$ Å) move to two sides of the figure. This movement results in a changing of phase transition point (cutoff). The smaller $W$ values are, the bigger cutoff are. When the $W$ is 0.2, the cutoff becomes big. At this time, the proportion of RNA is significant. It can be seen from the figure that the number of iRMSD $\leq 5$ Å is relatively small, and the binding modes are not very similar. This phenomenon may be explained as RNA secondary structure is not always conserved[61]. This result also suggests that both the similarity of protein and RNA are needed in identifying a good template like previous study[27]. Overall, correlation of the protein–RNA structural similarity with the binding mode depends on the way of combining the similarity

of monomers. For $W = 0.9, 0.8, 0.7, 0.6, 0.5, 0.4$, the transition 3D2D score 0.4, 0.45, 0.5, 0.6, 0.65, 0.7 were determined respectively.

**Benchmarking of PRIME-3D2D and the determination of weight**. In order to confirm the combination parameter of protein and RNA similarity score, a protein 3D structure/RNA secondary structure docking method was implemented in a program named PRIME-3D2D (3D/2D Protein–RNA Interaction ModEling). Figure 2a shows the outline of the approach. PRIME-3D2D was benchmarked on NRBC90 targets using NRBC349 as template library. For each target, docking models were generated by PRIME-3D2D and ranked by 3D2D score of several kind of weight combining TM-score and SSI. The result of benchmarking is shown in Supplementary Fig. 5. The weight is chosen as 0.8 for that the highest success rate of top 10 predictions, which is almost as good as PRIME[27].

Like previous study[27], the success rate of predicting "acceptable" model almost reaches the highest value at top four. Different weights result in various cutoffs (Fig. 1). But a higher cutoff value will lead to missing templates on which accepted models can be built (Supplementary Fig. 6). The curve labeled with "$W = 0.8$ with cutoff 0.45" is the best result. So, the weight was selected to 0.8. The success rate of top 10 is 0.63, which is just slightly smaller than the highest success rate (0.65) obtained by PRIME[27] and slightly better than that of PRIME2.0[62] (Fig. 3). This result suggests that RNA secondary structure is a strong constrain to reduce the RNA potential 3D conformation. This is a reason why RNA 3D structure prediction method can work by assembling 3D fragments which are collected by RNA secondary structure similarity[63]. And the subtle difference may be that the RNA comparison procedures are different. PRIME uses SARA, PRIME2.0 uses RMalign, and PRIME-3D2D uses LocARNA.

**Benchmarking of PRIME-3D2D**. Besides building models, we apply the PRIME-3D2D to predict the binding sites with 3D2D score in NRBC439. The cutoff (0.45) was applied to identify good templates for binding sites prediction.

In Fig. 4, it shows the results of binding sites prediction in NRBC439. Like building interact model in benchmark, the best MCC is reached at the top three prediction. For top 10 predictions, the MCC of binding sites prediction on protein and RNA are about 0.75 (Fig. 4c) and 0.70 (Fig. 4d), respectively. Comparing Fig. 4a, c (or Fig. 4b, d), the 3D2D score with cutoff 0.45 almost has detected all possible models. These results indicate that PRIME-3D2D with 3D2D score can be applied to predict binding sites.

**Comparison with other methods on PDB and genome scale data**. In order to evaluate our binding prediction approach PRIME-3D2D, we made a comparison with the current binding site prediction methods on PDB and genome scale data sets. The results show that PRIME-3D2D performed better than the current existing methods.

On PDB scale, PRIME-3D2D(PDB) was compared with the RNA-binding site predictors on independent testing sets (RB75, RB172, and RB344). These three independent testing sets and the results of other RNA-binding site predictors were grabbed from the review article[64], in which the authors benchmarked the softwares for predicting RNA-binding sites on proteins. From Fig. 2b, we can see that in the RB75 data set, the Meta-predictor has the highest AUC, and PRIME-3D2D achieves the best value among all the other evaluation indexes. In the data set RB172 (Fig. 2c), all evaluation indexes of PRIME-3D2D are better than other methods except ACC. In the RB344 data set (Fig. 2d),

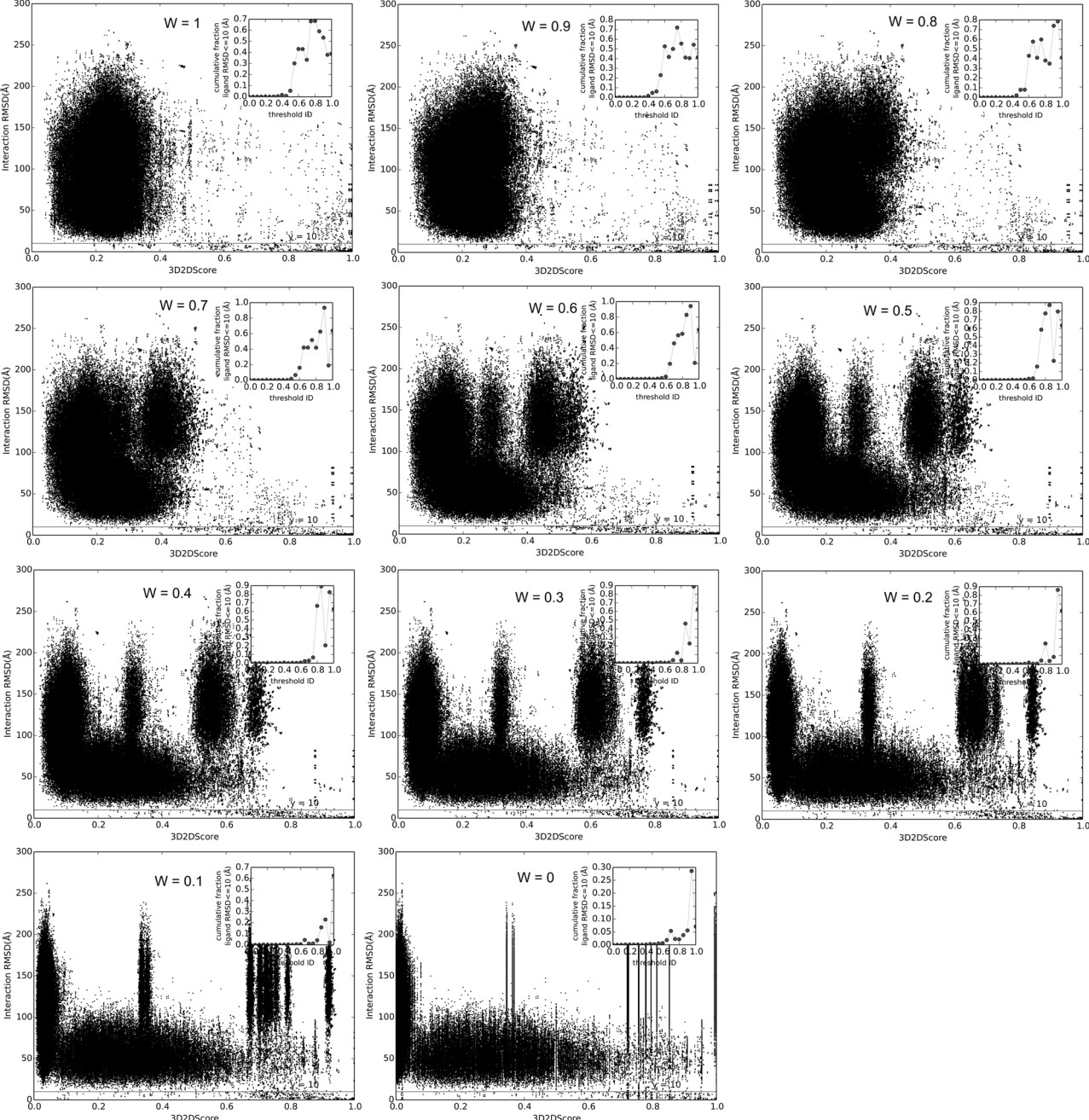

**Fig. 1 Binding modes vs structural similarity within different weights.** iRMSD is plotted against 3D2D score (W * TM-score + (1 − W) * RNA secondary structure identity), in all-to-all comparison of NRBC439. The accumulative fraction of iRMSD ≤ 5 Å is defined as that the number of pairs with the iRMSD ≤ 5 Å is divided by all the pairs within one bin. The transition point is defined as that the similarity score threshold with which the accumulative fraction of the iRMSD ≤ 5 Å begins changing from 0 to non-zero value. The insets show the fraction of complex pairs with iRMSD ≤ 5 Å is plotted in 0.05 bins to show the phase transition. The subfigures correspond to various W.

PRIME-3D2D reaches the lowest SP, but obtains the highest values in the remaining evaluation indexes. Taken together, PRIME-3D2D achieves the best results on these three data sets by using MCC as the main indicator and others as auxiliary evaluation indexes. As the above methods except PRIME-3D2D were developed to predict the binding site of RNA on proteins, we compared the performance of PRIME-3D2D and PRIME2.0 on prediction the binding sites (Tables 1 and 2). The binding sites of PRIME2.0 were calculated from the predicted complex structure model. The interface residue was defined by <4.5 Å distance between any heavy atom of the protein and any heavy atom of the

RNA. And the binding sites of PRIME-3D2D were predicted based on the alignment of target and template (Fig. 2a). In Table 1, it shows the result of the PRIME2.0/PRIME-3D2D for protein-binding site prediction. we can see that on the RB75 and RB344 data sets, PRIME-3D2D is superior to PRIME2.0 in protein-binding sites prediction. It maybe illustrates that the secondary structure of RNA is helpful to find more accurate binding sites. Table 2 shows the result of the PRIME2.0/PRIME-3D2D for RNA-binding site prediction. From Table 2, we can see that PRIME2.0 is superior to PRIME-3D2D in RNA-binding sites prediction. It maybe illustrates that the three-dimensional

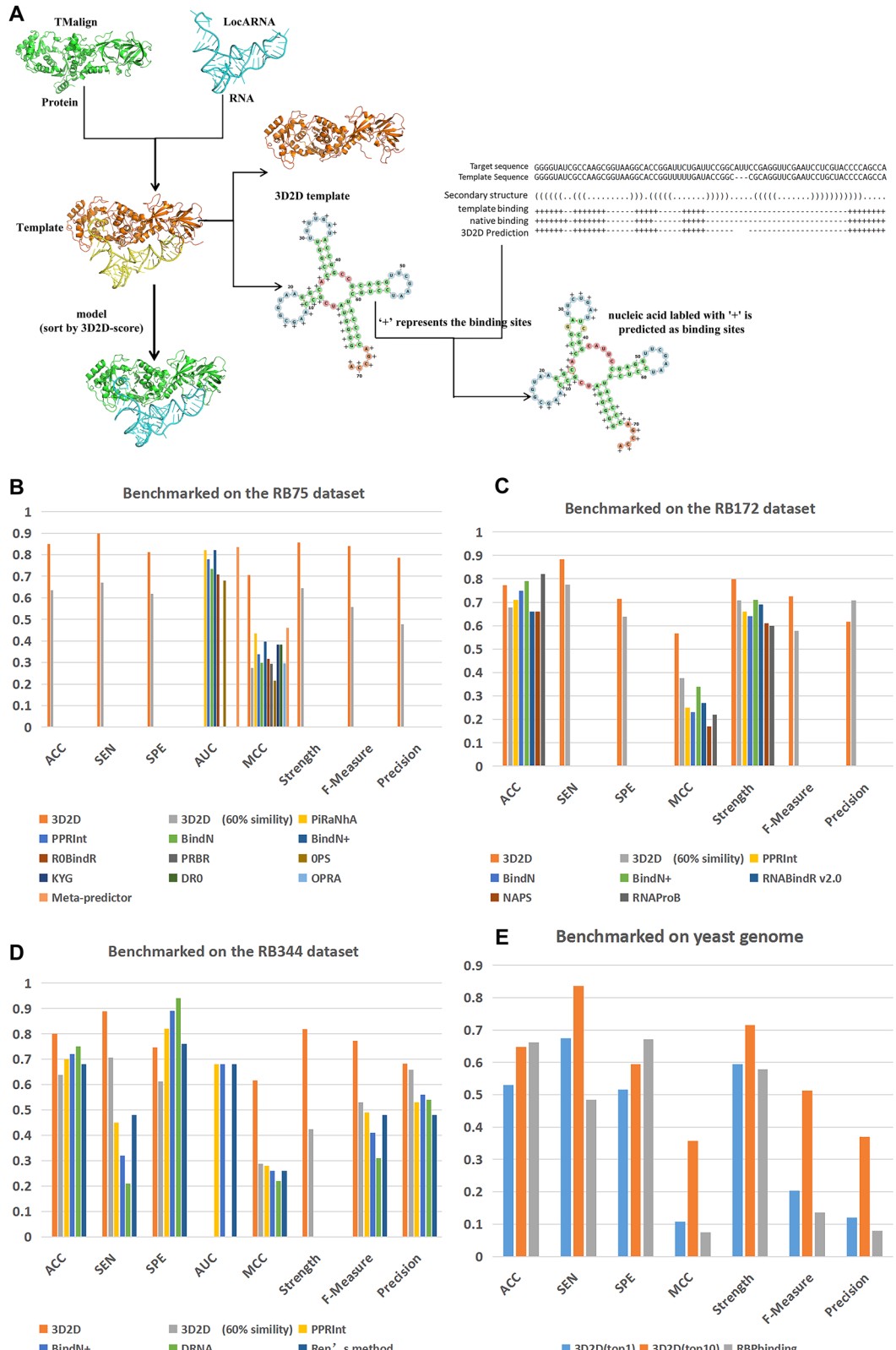

complex structure of RNA is better than that of the secondary structure to find more accurate binding sites on protein.

On genome wide, PRIME-3D2D (genome) predicted the binding sites on RNA of yeast transcriptome, but the MCC is lower than that in PDB wide. At first, RNAs of the yeast transcriptome are employed as the targets and NRBC439 is used as the templates. However, PRIME-3D2D does not work well

(Supplementary Fig. 7). This result may be caused by the lacking good templates. So, the PDB-based template library is extended to yeast genome wide. In this case, all target-RNA interaction pairs are used as templates. But in the stage of searching the template, the target itself will be excluded. As a result, the MCC of PRIME-3D2D is 0.357 and the ACC is 0.647 for top 10 (Fig. 2e). Then, PRIME-3D2D is compared with RBPbinding[46] on yeast

**Fig. 2 The procedure of the PRIME-3D2D and the result of PRIME-3D2D comparing with other methods. a** is the schematic diagram of the PRIME-3D2D. The input protein and RNA structures are aligned to the templates by TM-align and LocARNA, respectively. The models of the complex are sorted by the 3D2D score (see text). In binding site prediction, the 3D3D model is converted to a 3D2D model as a template (protein maintains 3D structure, RNA maintains 2D structure). If the base (residue of the protein) in the target RNA is aligned to the binding site of the template RNA (the binding site of the protein), then this base (residue) is predicted to be the binding site. **b–d** show the results of PRIME-3D2D (top 1) comparing with the state-of-the-art methods for predicting RNA-binding site in protein on three PDB data sets (RB75, RB172,and RB344, respectively). In addition to the PRIME-3D2D results, others are from ref. [64]. 60% similarity indicates that the similarity between target and template is within 60%. **e** shows the result of PRIME-3D2D comparing with RBPbinding[46] for predicting of protein-binding site on RNA on the yeast genome. These results show that PRIME-3D2D is better than other methods in predicting binding sites.

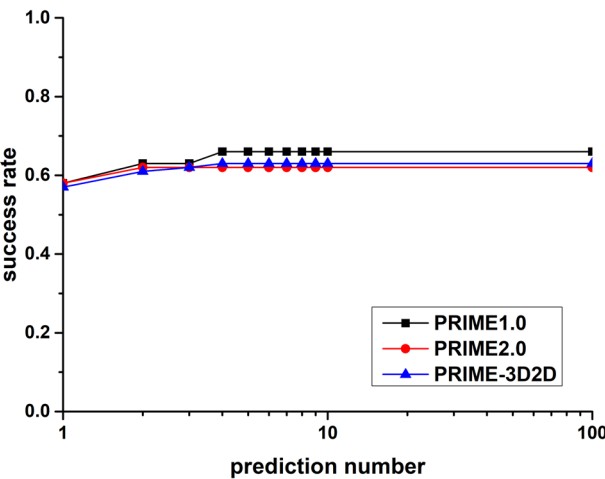

**Fig. 3 Comparison of the success rate of PRIME-3D2D/PRIME1.0/ PRIME2.0 on the NRBC439 data set.** Targets (90 newer complexes) were predicted using templates (349 older complexes). The models are ranked by 3D2D score and structural score in PRIME-3D2D and PRIME1.0/PRIME/ 2.0, respectively. The docking of a complex was successful if at least one prediction within a set number of predictions was successful. *X* axis stands for the top n of the predicted number.

transcriptome(Fig. 2e). RBPbinding was better than PRIME-3D2D in the terms of ACC and SP, but PRIME-3D2D achieved higher values in terms of SN, MCC, strength, F-Measure, and PRE. An example is shown in Fig. 5. In summary, the results show that PRIME-3D2D performed better than the current existing methods on PDB wide and genome wide. In the yeast genome, regardless of PRIME-3D2D or RBPbinding, the MCC values are not particularly high, indicating that there is great room for improvement in the protein-binding sites prediction on RNA. The accuracy of PRIME-3D2D is reduced when the template and target are de-redundant by 60% sequence similarity.

## Conclusion
PRIME-3D2D is as good as PRIME in building models on NRBC439 benchmark. For further expanding the application of PRIME-3D2D, we employ the model to predict protein or RNA-binding sites on PDB wide. Testing on NRBC439 shows this model achieves the MCC of 0.7. Subsequently, we perform yeast protein–RNA interactome vs NRBC439 to predict the binding sites on RNA. However, the result shows that more protein–RNA interactions and binding sites should be added to the template library. So, the yeast interactome was used as template to predict binding sites. In this process, we use the secondary structure of the RNA predicted by SeqFold, which is based on experimental data, and assume that the secondary structure of RNA before and after binding RBP has not been changed to simplify the model. The results show that PRIME-3D2D can be used to predict

binding sites on genome wide. Comparison with the state-of-the-art methods shows PRIME-3D2D outperformed on both PDB and genome wide data.

## Methods
**RNA2dA.** In order to test the effect of RNA sequence and secondary structure in searching protein–RNA complex templates, an RNA alignment method considering RNA sequence with secondary structure is needed. Beagle[58] is a proper alignment approach for RNA alignment. But the software is not open to access. So, we implement a similar approach called RNA2dA. Similar to the Beagle, the BEAR coding[56] was chosen as the representation of RNA secondary structure. Needleman–Wunsch algorithm[65] was employed to accomplish the global alignment. Then we built a score matrix similar to Blocks Substitution Matrix(BLOSUM)[66] according to the blocks from Rfam[67]. In the sequence alignment procedure, the scoring matrix of RNA2dA is different with Beagle's substitution matrix (Matrix of Bear-encoded RNAs), which used Percent Accepted Mutations (PAM). The scoring matrixes of RNA2dA were generated with different weights by combining RNA sequence similarity and RNA secondary structure similarity. Finally RNA2dA was benchmarked on BRAliBase II[57].

*BEAR encoding.* BEAR is a representation of RNA secondary structure which is first introduced in ref. [56]. The BEAR representation is same as Beagle.

*RNA blocks from Rfam.* Seed alignments of Rfam[67] are used to construct RNA blocks. RNA block is defined as multiple sequence alignment fragments with the length of alignment greater than five nucleotide without insertions and deletions. The RNA sequences of blocks are extracted from the multiple sequence alignment. And the secondary structures are extracted from the lines labeled "SS_cons" in seed alignment of Rfam. Then RNA is converted to BEAR encoding with BEAR software. After these steps, we obtain the blocks in BEAR coding.

*RNABLOSUM matrix.* The C source code[66] of generating protein BLOSUM matrix is modified to calculate the RNABLOSUM matrix at first. Redundancy of RNA with BEAR encoding are then removed with different RNA secondary structure identity (SSI) (with BEAR representation). With different cutoffs x, various RNABLOSUM matrixes labeled with RNABLOSUMx are calculated.

*Scoring matrix of RNA2dA.* In order to consider RNA sequence information, we combine the RNABLOSUMx matrix and NUC.4.4 matrix as follows:
    Score (i, j) = RNABLOSUMx (BEAR (i), BEAR (j)) + NUC.4.4 (i, j) * bonus
    Score (i,j) is the scoring matrix of RNA2dA. BEAR (i) and BEAR (j) are the BEAR representation of nucleotide i and nucleotide j. NUC.4.4 is the scoring matrix for RNA sequence. The matched nucleotides will be scored to 5 and the mismatched nucleotides will be scored to −4. NUC.4.4 was downloaded from ftp://ftp.ncbi.nih.gov/blast/matrices/. The bonus indicates the effect of RNA sequence in RNA2dA. If the bonus is set as 0, RNA2dA will only use RNA secondary structure to align RNA. In RNA2dA, the gap open penalty is set to 10. And the gap extend penalty is set to 2. The alignment is accomplished by Needleman–Wunsch algorithm.

*Benchmark of RNA2dA and comparison with LocARNA.* RNA2dA was benchmarked and compared with LocARNA on BraliBase II[57]. Like dealing alignments in Rfam, sequence and secondary structure are extracted from the BraliBase II. Sum-of-pairs (SPS), that is defined as the fraction of correctly aligned nucleotide pairs within one alignment is used as quality measurement.

**The difference between PRIME and PRIME-3D2D.** The differences between PRIME and PRIME-3D2D are mainly in: 1. PRIME requires the 3D structure of the RNA as input, whereas PRIME-3D2D requires the 2D structure of the RNA; 2. PRIME is used for prediction complex structure. When applied to the PDB scale, the PRIME-3D2D model can predict both the complex structure and the binding site; on the genome scale, PRIME-3D2D is used to predict the binding site of proteins on RNA; 3. When searching templates in RNA, the algorithm is different. SARA/RMalign is used for RNA alignment in PRIME1.0/PRIME2.0, and

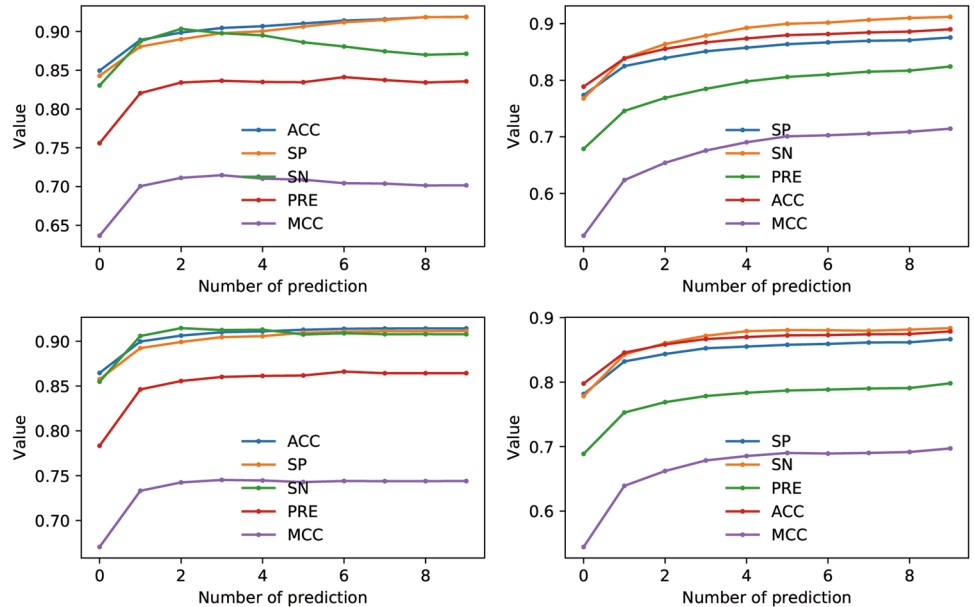

**Fig. 4 RNA and protein-binding sites prediction on NRBC439.** SP, SN, PRE, ACC, and MCC are plotted against number of predictions, in all-to-all comparison of NRBC439. The 3D2D-models are sorted by 3D2D score. For all target, we calculate the mean value of best models on top n prediction. Subfigures **a**, **c** are plotted for RNA-binding sites prediction. Subfigures **b**, **d** are plotted for protein-binding sites prediction in RNA sequence. The templates are filtered out with the 3D2D score under 0.45 are plotted at the two bottom subfigures.

**Table 1 Performance of the PRIME2.0/PRIME-3D2D for protein-binding site prediction.**

| Method | Data set | Performance (binding site on RNA) | | | | | | |
|---|---|---|---|---|---|---|---|---|
| | | ACC | SN | SP | MCC | Strength | F-measure | PRE |
| PRIME-3D2D | RB75 | 0.849 | **0.965** | 0.778 | **0.721** | **0.872** | **0.828** | **0.725** |
| | RB172 | 0.694 | **0.769** | 0.651 | 0.404 | 0.710 | **0.647** | 0.559 |
| | RB344 | 0.762 | **0.814** | 0.725 | **0.532** | **0.770** | **0.742** | **0.682** |
| PRIME2.0 | RB75 | **0.940** | 0.531 | **0.972** | 0.532 | 0.752 | 0.563 | 0.599 |
| | RB172 | **0.781** | 0.546 | **0.889** | **0.467** | **0.717** | 0.610 | **0.692** |
| | RB344 | **0.840** | 0.599 | **0.903** | 0.509 | 0.751 | 0.610 | 0.621 |

Bold values indicate that the method performs better in RB75/RB172/RB344.

**Table 2 Performance of the PRIME2.0/PRIME-3D2D for RNA-binding site prediction.**

| Method | Data set | Performance (binding site on protein) | | | | | | |
|---|---|---|---|---|---|---|---|---|
| | | ACC | SN | SP | MCC | Strength | F-measure | PRE |
| PRIME-3D2D | RB75 | 0.850 | **0.90** | 0.811 | 0.706 | 0.856 | 0.840 | 0.787 |
| | RB172 | 0.772 | **0.884** | 0.714 | 0.567 | 0.799 | **0.726** | 0.616 |
| | RB344 | 0.801 | **0.889** | 0.747 | 0.617 | 0.818 | 0.772 | 0.682 |
| PRIME2.0 | RB75 | **0.918** | 0.876 | **0.938** | **0.812** | **0.907** | **0.872** | **0.869** |
| | RB172 | **0.900** | 0.687 | **0.946** | **0.650** | **0.817** | 0.710 | **0.734** |
| | RB344 | **0.911** | 0.806 | **0.940** | **0.742** | **0.873** | **0.799** | **0.792** |

Bold values indicate that the method performs better in RB75/RB172/RB344.

LocARNA is used for structural alignment in PRIME-3D2D; 4. The finally complex structure scores are different. PRIME1.0/2.0 use the lower score between protein and RNA as the complex structure score, but PRIME-3D2D employs a linear combination between protein and RNA structure scores.

**Protein–RNA interaction and binding sites data**. NRBC439[27] was downloaded from http://www.rnabinding.com/PRIME.html. NRBC439 is used as the template library, in which all-to-all alignment of protein–RNA complex structure is conducted like PRIME[27]. To compare LocARNA with RNA2dA and predict the new complex structure based on the known complex structures, NRBC439 is split into two parts. 80% with an older deposit date are designated as the templates (NRBC349), and 20% with a newer deposit date are designated as targets (NRBC90), which is consistent with the division method in the PRIME1.0 and PRIME2.0. The binding site in NRBC439 is defined by distance ≤4.5 Å between any heavy atom of the protein and any heavy atom of the RNA.

To obtain the RNA secondary structures, at first, we downloaded the sequence file in FASTA format of *Saccharomyces cerevisiae* (*S. cerevisiae*) from http://ouyanglab.jax.org/seqfold/instructions.html#Prerequisites. We predicted the RNA secondary structure by following the tutorial of SeqFold[68], which converts the high-throughput RNA structure information to structure preference profile (SPP). After getting the SPP file, SeqFold will determine that the base is on a single-strand or double-strand, then the RNA secondary structure will be predicted. The

| | target | GBP2 (protein) | YKL068W-A(RNA) | template | TMscore | RNAsimilarity | 3D2DScore | TP | FP | TN | FN | N |
|---|---|---|---|---|---|---|---|---|---|---|---|---|
| | template | Npl3 (protein) | YNL113W (RNA) | Npl3YNL113W | 0.85135 | 0.47923 | 0.77693 | 0.952 | 0 | 0.958 | 0.002 | 0.004 |

```
>YKL068W-A
native   sequence:CUAGUAACUCAUAAAUCAACGACAUUAUAUAUCAUCAACUACAAUGCAGGCAAACCAUUCCGUCAGUUACCUUUACGAAUCAAGUACUUCAAAGAGGUCAAAUGGGCUCUUUUCU
CAAACACAAAAGCAAGGAAGUUUUCAAAAAGCUUUAAGUCAAAUAGCACAGGAAGAAAUAGAAGAUGAAGACCUGAAUGGUAGACCUCAACACAGGAUCUUUAAACUCCCGUUAAGUUAAAAUAUUUGGACUCAAAUGAG
UGCUAUGACGGAAAAAAUUUGGUAAACUGUGUAGAGCCAAUUUUUUGGCGGUAAAUUUCGUUUAGCUGUUUUAUUUUCACAUUUUUAAUUGUCAAUGCUGGAGGUUGUUAAAGUUUUGGCUAGCUGUUCUGAACAAUCCUU
GUAUACCGGUGAGACUUUUUUGUUAUAGAUAUUGGCGAUAUUCAUUUUUUUAGGAAGGUGUUUUUUAUAGAUUUUUCAAUUUUUUUUUGUGUAGAUGCAUGUUUUGAUUACUUAUGCAUAGUACGAAUUUUUUAGUUUA
UUAAGCUAUAGAGGUUUUUUUUUUUUUCACCUCUUUACCCACAAAUUUAAUAGCUAUUGUGUAGAAUAACUUUUAAUGGAAAACAUUUGCAGUAUUUUA
native binding site:-------------------------------------------------------------------------------+++++++++++++++++++-------------
-----------------------------------------------------------------------------------------------------------------------------------------
----------------------------------------------------------------------------------------------------------------------------------------
------------------------------------------------------------------------------------------------------------------------------------
-------------------------------------------------------------------------------------
3D2D_PRIME predice:  ------  -----------  --------  ---------------------------------------------+++ +++++++++++++++++-------------
---------------------------------------------------------------------------------------------------------------------------------------
---  ------------------  --------------  -----------------------------------------------------------------------------------------
------------------------------------  -------------------------------------------------
RNAbinding predice:------------------+++++++++++++++++++++++++++++++++--------------------++++++++++++++++++++++--++++++++
+++++++++++++++++++++++++++++++++++++++++++++++++++++++-.+++++++++++++++++++++++++-----.++++++++++++++++++++++++++++++++++++++++++++-
---------------------------------------------------------------------------------------------------------++++++++++++++
++++++++++++-------------++++++++++++++++++++-------------------------------------------------
++++++++++++-------------+++++++++++++++++++----------------------------------------------------
```

**Fig. 5 PRIME-3D2D and RBPbinding predict the binding sites of the protein (GBP2) on RNA (YKL068W-A).** In PRIME-3D2D, the target protein GBP2 uses Npl3 as the template and the target RNA YKL068W-A employs YNL113W as the template. The top right corner shows the scores between target and template. In addition, the sequence of RNA YKL068W-A, the natural binding sites, the PRIME-3D2D prediction result and the RBPbinding prediction result on target RNA are shown. From the prediction results, it is not difficult to find that the false positive of RBPbinding is higher in this example.

predicted secondary structure of each transcript was in CT format, so we then used the RNA structure[69] to convert the CT format to dot-bracket format. We assume that the secondary structural state of RNA remains the same before and after binding to the protein. To obtain the native protein-binding sites file, we downloaded the BED file corresponding to the 20 yeast proteins from CLIPdb[13], which include the information of protein-binding sites on RNA, and the 3D structures of these 20 proteins were downloaded from PDB website and ModBase database[70]. The CDS file of S.cerevisiae was downloaded from the Ensemble database. If the binding site is included in the BED file and included in the CDS file, we take out the transcripts, respectively. According to the BED and the transcript files, the native binding sites files were defined.

**Target/template alignment and similarity score**. NRBC90 vs NRBC349 pairwise alignments were performed by two approaches to detect the ability of templates searching. The first approach is LocARNA, which combines RNA sequence and RNA secondary structure information to align RNAs. It was developed for RNA local or global structural comparison. It is currently one of the most effective and convenient methods for RNA secondary structure alignment. This method can classify RNAs according to RNA secondary structure. Hence, this method can be used to align the secondary structure of the target RNA to the known secondary structure of RNA, to find a known RNA similar to the target. The parameter of LocARNA was set to run global alignment. The second approach is RNA2dA (bonus is set to 0). For the structural alignment of protein, we chose TM-align following the previous study[27]. The interaction RMSD (iRMSD)[59] is used to characterize the binding mode of complexes of different monomers.

In NRBC439 set, all-to-all pairwise alignments were performed by LocARNA and TM-align for RNAs and proteins, respectively. In order to get a normalized score to measure the RNA similarity, we define the SSI like sequence identity. The similarity of the complex is defined as the linear combination of TM-score and SSI, called 3D2D score ($W *$ TM-score $+ (1 − W) *$ RNA SSI). The combination parameter (weight) is optimized in benchmarking.

**Building and evaluating protein–RNA 3D structure based on 3D2D alignment**. After a good template is selected, the target protein is superimposed on the template protein by TM-align. Then the RNA between template and target is aligned by LocARNA. At the same time, the transition matrix is generated by minimizing the RMSD of aligned nucleic acid (C3' atoms).

After the model is built, the ligand RMSD (RMSD of RNA C3' atoms) between the model and the native structure is calculated. The quality of the model is measured by ligand RMSD. Like PRIME1.0 and PRIME2.0, an "acceptable" prediction is defined as the model with ligand RMSD ≤10 Å.

**Binding sites predicting using protein–RNA PRIME-3D2D**. In addition to building the 3D structural model of protein–RNA interaction, PRIME-3D2D can be used to predict binding sites in both protein and RNA sequences on PDB scale and predict binding sites in RNA sequence on genome wide (Fig. 2a). Predicting binding sites in the target depends on the templates. For each case, nucleotides of target aligned to binding sites in the template are predicted as binding sites, and that aligned to other nucleotides are predicted non-binding sites.

The performance in binding sites prediction is evaluated by sensitivity (SN), specificity (SP), precision (PRE), accuracy (ACC), Matthews Correlation Coefficient (MCC), Strength, F-measure, and AUC, which are defined as follows:

$$\text{Sensitivity (SN)} = TP/(TP + FN)$$
$$\text{Specificity (SP)} = TN/(TN + FP)$$
$$\text{Precision (PRE)} = TP/(TP + FP)$$
$$\text{Accuracy (ACC)} = (TP + TN)/(TP + TN + FN + FP)$$
$$\text{Matthews Correlation Coefficient (MCC)} = (TP*TN − FP*FN)/$$
$$\sqrt{(TP + FN)*(TP + FP)*(TN + FP)*(TN + FN)}$$
$$\text{Strength} = (SN + SP)/2$$
$$F − \text{measure} = (2*PRE*SN)/(PRE + SN)$$

where, TP is true positive, FN is false negative, TN refers to true negative, and FP refers to false positive. AUC stands for the area under the receiver operating characteristic curve.

**Reporting summary**. Further information on research design is available in the Nature Research Reporting Summary linked to this article.

## Data availability

PDB data. The benchmarked data NRBC439 was downloaded from http://www.rnabinding.com/PRIME.html. And the independent test set RB75, RB172 and RB344 were downloaded from https://www.sciencedirect.com/science/article/pii/S1047847711002851?via%3Dihub, https://www.ncbi.nlm.nih.gov/pmc/articles/PMC3962366/#!po=5.00000, and https://bmcbioinformatics.biomedcentral.com/articles/10.1186/s12859-015-0691-0/tables/4, respectively.

Genome data: The CLIP-seq data were downloaded from CLIPdb. And the experimentally determined RNA secondary structure data were download from https://www.ncbi.nlm.nih.gov/pubmed/23064747.

BRAliBase II was downloaded from https://www.ncbi.nlm.nih.gov/pubmed/15860779.

Any remaining data can be obtained from the corresponding author upon reasonable request.

## Code availability

PRIME-3D2D is user-friendly and freely available at http://www.rnabinding.com/PRIME-3D2D/.

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

## Acknowledgements

We thank the National Supercomputer Center in Guangzhou for the support of computing resources. National Natural Science Foundation of China [31100522]; National High Technology Research and Development Program of China [2012AA020402]; the Fundamental Research Funds for the Central Universities [2016YXMS017]; Special Program for Applied Research on Super Computation of the NSFC-Guangdong Joint Fund (the second phase) [U1501501]. Funding for open access charge: Fundamental Research Funds for the Central Universities [2016YXMS017]. PRIME-3D2D is freely available at http://www.rnabinding.com/PRIME-3D2D/.

## Author contributions

J.F.Z. and J.X. developed the PRIME-3D2D and the webserver. J.X., J.F.Z., X.H., X.X.T., and S.Y.L. wrote, reviewed, and edited the manuscript.

## Competing interests

The authors declare no competing interests.
