## [Peer Review File · Communications Biology]

Reviewers' comments:

Reviewer #1 (Remarks to the Author):

Xie and coauthors report a new computational method PRIME-3D2D to predict binding sites of protein-RNA interaction and model these complexes. This method represents a progress from their previous version PRIME (3D3D), which relies on RNA secondary structure and sequence similarity between the target and template RNA to predict binding sites of protein-RNA interaction. This study is of general interest to the field of protein-RNA interactions and structural biology.

Several comments are listed below for consideration:

(1) On page 5, line 108, "In conclusion, PRIME-3D2D is as good as PRIME in building models on NRBC439 benchmark".

This comparison between PRIME-3D2D and PRIME should be shown in Figure 1 as well along with other state-of-art methods to allow assessment of the advance of the current method from prior version.

(2) In Figure 1, page 8.

I am concerned why AUC is not presented or calculated for the current PRIME-3D2D method to allow comparison with other methods.

(3) In Figure 1, page 8

It appears there is a general trend of outperforming and underperforming of PRIME-3D2D over other methods, in different ways to assess the performance. In particular, PRIME-3D2D appears to outperform in all datasets in MCC, precision and SEN. This indicates PRIME-3D2D may be good at prediction of TP (true positives) but not TN (true negatives). If this is true, then a discussion of the underlying cause may be helpful for improving the current algorithm for better performance. And a general conclusion about current method should mention both outperformance and underperformance depending on the category of statistics.

See the apparent trend below (according to Figure 1):

SEN is short for Sensitivity = $n_{TP} / (n_{TP} + n_{FN})$, where TP means true positive and FN means false negative predictions). Precision = $n_{TP} / (n_{TP} + n_{FP})$.

However, it underperforms in SPE (Specificity = $n_{of_TN} / (n_{of_TN} + n_{of_FP})$) as well as in certain cases ACC (Accuracy = $(n_{of_TP} + n_{of_TN}) / n_{all_predictions}$).

Nonetheless, it seems PRIME-3D2D outperforms in MCC, which is a type of Matthews Correlation Coefficient to evaluate the overall performance, taken into account both positive and negative predictions.

(4) Page 5 line 107, "The accuracy of 3D2D-PRIME is reduced when the template and target are de-redundant by 60% sequence similarity, indicating that some proteins may bind to similar RNAs".

It's not surprising when sequence similarity cutoff affects accuracy of predictions because more high confidence templates may be excluded. Not sure the explanation of "some proteins may bind to

similar RNAs" is very informative here.

(5) Minor typos, etc and suggested corrections

Page 3 line 65, "is depended on" -> "depends on"

Page 4 line 83, "expect ACC" -> "except"

Figure 1 legend, first sentence. "..and the result of PRIME-3D3D". Is it meant for "PRIME-3D2D"?

Reviewer #2 (Remarks to the Author):

The authors present a novel approach, PRIME-3D2D, to predict protein-RNA interaction considering protein 3D structure and RNA 2D structure. This method was applied to the binding sites at a transcriptome wide level. It would provide a possible way to study protein-RNA interactions at a large scale. However, some concerns need to be addressed.

Major Points:

1. The paper does not read like a Correspondence. It should be reformat as a research article and more detailed Results/Methods need to be provided carefully.
2. RNA secondary structure will be affected by different protein binding at different conditions/cells. Is it reasonable to predict RNA-protein binding derived from CLIPdb using structure information derived from SeqFold?
3. PRIME-3D2D integrates RNA secondary structure and RBP binding site data of yeast genome. However, a large amount of RNA secondary structure and RBP binding site data of other species have been curated and available in public databases. The authors need to apply this method to more species and more data. For instance, the RBP binding sites were collected from a subset of CLIPdb. More updated CLIPdb/POSTAR2 datasets need to be included. And more details about how to integrate RBP binding sites need to be described.
4. The authors mentioned that PRIME-3D2D was derived from the protein-RNA 3d3d-model. I understand that Protein-RNA 3D3D-model can't be used on genome wide for the lack of RNA 3D structures. However, for those data with 3D structures, the authors should compare and show the performances of both PRIME-3D2D and protein-RNA 3D3D.
5. The authors mentioned two approaches detecting the ability of template searching, LocARNA and RNA2dA. The authors gave a detailed introduction to RNA2dA, but not to LocARNA.
6. Interaction RMSD (iRMSD) and 3D2D-score are two important index values of PRIME-3D2D. The similarity of binding mode between the target and the template is measured by interaction RMSD (iRMSD) and 3D2D-score. In Supplementary Fig. 5, the authors show that iRMSD is plotted against 3D2D-score. Are these two indicators the same? What is the relationship of them?
7. Figure 1.B, C and D show the results of PRIME-3D2D (top 1) comparing with the state-of-the-art methods for predicting RNA-binding site in protein on three PDB datasets (RB75, RB172 and RB344, respectively). The state-of-the-art methods for predicting RNA-binding site in protein, like PiRaNha, BindN and so on, should be described in the Methods. And more details need to be provided in the results. For instance, why are the control algorithms applied differently in RB75, RB172 and RB344? Why do the control algorithms have different degrees of vacancy in SN, SP, PRE, ACC, Strength, F-

measure and AUC? It is known from the Figure 1.B, C and D that MCC is the evaluation index that can be calculated by all algorithms. Why not use it as the optimal evaluation index?

Minor Points:

We found some typos in this manuscript. Some examples are listed here:

8. In page 5 of manuscript, "In summary" does not need to be bold.
9. In page 1 of supplementary materials and methods, "interactionome" should be "interactome".
10. In page 8 of supplementary materials and methods, "Figule" should be "Figure".

Reviewer #3 (Remarks to the Author):

1. In RNA2dA, the author used the matrix PAM, not BLOSUM, which is different from the Beagle method. Please explain the difference between them and why choose PAM.
2. The author have developed a tool of RNA-binding site prediction, named PRIME, which needs the information of protein 3D structure and RNA 3D structure. The novel method named PRIME-3D2D (protein 3D structure and RNA 2D structure needed) seemed less input information? The reader wonder to know if there are any difference or advance from PRIME? Please explain it in detail.
3. NRBC439 is split into two parts. 80% with an older deposit date are designated as the templates (NRBC349), and 20% with a newer deposit date are designated as targets (NRBC90). The rule of deciding which data is template and which data is target is important. Please tell the rule. If necessary, cross-validation should be used here.
4. I submitted a task in the website of PRIME-3D2D, but unfortunately I did not receive the result. Please check whether the website works.
5. NRBC439, as the main dataset in this work, I think it is necessary to divide the data into train data and test data using 10-fold cross-validation, in order to ensure the reliability of the prediction results.

Reviewers' comments:

Reviewer #1 (Remarks to the Author):

Xie and coauthors report a new computational method PRIME-3D2D to predict binding sites of protein-RNA interaction and model these complexes. This method represents a progress from their previous version PRIME (3D3D), which relies on RNA secondary structure and sequence similarity between the target and template RNA to predict binding sites of protein-RNA interaction. This study is of general interest to the field of protein-RNA interactions and structural biology.

Several comments are listed below for consideration:

Q: (1) On page 5, line 108, "In conclusion, PRIME-3D2D is as good as PRIME in building models on NRBC439 benchmark".

This comparison between PRIME-3D2D and PRIME should be shown in Figure 1 as well along with other state-of-art methods to allow assessment of the advance of the current method from prior version.

A: Thanks for the suggestion of reviewer, we made the following comparison between PRIME2.0 and PRIME-3D2D. 1, On NRBC439 data set, we compared them in complex structure modeling; 2, On the data sets of RB75/RB172/RB344, we compared them in the performance of binding sites prediction. The results are as follows:

Figure 3. The result of PRIME1.0, PRIME2.0 and PRIME-3D2D on modeling complexes on NRBC439 dataset. The result showed that PRIME-3D2D/PRIME/PRIME2.0 get relatively close accuracy when constructing three-dimensional complex structures. The subtle difference may be that the RNA comparison procedures are different. PRIME uses SARA, PRIME2.0 uses RAlign, and PRIME-3D2D uses LocARNA. We put it in the lines 189-191 and the figure 3 of the revised version.

The binding sites of PRIME2.0 were calculated from the predicted complex structure model. The interface residue was defined by $< 4.5 \text{ \AA}$ distance between any heavy atom of the protein and any heavy atom of the RNA. And the binding sites of PRIME-3D2D were predicted based on the alignment of target and template (Figure 2A in the revised version). Table1 and Table2 are the comparison results of PRIME2.0 and PRIME-3D2D on binding site on RNA and binding site on protein prediction, respectively.

Table 1. Performance of the PRIME2.0/PRIME-3D2D for protein-binding site prediction

Method	Data Set	Performance (binding site on RNA)						
		ACC	SN	SP	MCC	Strength	F-Measure	PRE
PRIME-3D2D	RB75	0.849	0.965	0.778	0.721	0.872	0.828	0.725
	RB172	0.694	0.769	0.651	0.404	0.710	0.647	0.559
	RB344	0.762	0.814	0.725	0.532	0.770	0.742	0.682
PRIME2.0	RB75	0.940	0.531	0.972	0.532	0.752	0.563	0.599
	RB172	0.781	0.546	0.889	0.467	0.717	0.610	0.692
	RB344	0.840	0.599	0.903	0.509	0.751	0.610	0.621

From table 1, we can see that on the RB75 and RB344 datasets, PRIME-3D2D is superior to PRIME2.0 in protein-binding sites prediction. It illustrated that the secondary structure of RNA is helpful to find more accurate binding sites in protein-binding site prediction.

Table 2. Performance of the PRIME2.0/PRIME-3D2D for RNA-binding site prediction

Method	Data Set	Performance (binding site on protein)						
		ACC	SN	SP	MCC	Strength	F-Measure	PRE
PRIME-3D2D	RB75	0.850	0.90	0.811	0.706	0.856	0.840	0.787
	RB172	0.772	0.884	0.714	0.567	0.799	0.726	0.616
	RB344	0.801	0.889	0.747	0.617	0.818	0.772	0.682

	RB75	0.918	0.876	0.938	0.812	0.907	0.872	0.869
PRIME2.0	RB172	0.900	0.687	0.946	0.650	0.817	0.710	0.734
	RB344	0.911	0.806	0.940	0.742	0.873	0.799	0.792

From table 2, we can see that PRIME2.0 is superior to PRIME-3D2D in RNA-binding sites prediction. It illustrated that the three-dimensional structure of RNA is better than that of the secondary structure to find more accurate binding sites on protein.

We put the table 1 and table 2 on the revised version and the introduction is placed on lines 220-235 of the revised version.

Q: (2) In Figure 1, page 8.

I am concerned why AUC is not presented or calculated for the current PRIME-3D2D method to allow comparison with other methods.

A: We appreciate the reviewer to figure out this important point. Calculation of AUC needed to determine different thresholds based on score. But in PRIME-3D2D, when the 3D2D-score is greater than or equal to 0.45, it is considered that the target has searched the appropriate template, and then PRIME-3D2D will predict the binding site. Hence, AUC is not able to measure the performance of PRIME-3D2D.

Q: (3) In Figure 1, page 8

It appears there is a general trend of outperforming and underperforming of PRIME-3D2D over other methods, in different ways to assess the performance. Specially, PRIME-3D2D appeared to outperform in all datasets in MCC, precision and SEN. This indicated PRIME-3D2D be good at prediction of TP (true positives) but not TN (true negatives). If this is true, then a discussion of the underlying cause may be helpful for improving the current algorithm for better performance. And a general conclusion about current method should mention both outperformance and underperformance depending on the category of statistics.

See the apparent trend below (according to Figure 1):

SEN is short for Sensitivity = $n_{TP} / (n_{TP} + n_{FN})$, where TP means true positive and FN means false negative predictions). Precision = $n_{TP} / (n_{TP} + n_{FP})$.

However, it underperforms in SPE (Specificity = $n_{of_TN} / (n_{of_TN} + n_{of_FP})$) as well as in certain cases ACC (Accuracy = $(n_{of_TP} + n_{of_TN}) / n_{all_predictions}$).

Nonetheless, it seems PRIME-3D2D outperforms in MCC, which is a type of Matthews Correlation Coefficient to evaluate the overall performance, taken

into account both positive and negative predictions.

A: Thanks for your comment. In the table below, we listed the values of TP, TN, FP, and FN on different data sets:

	RB75	RB172	RB344	Yeast genome(top1/top10)
TP	0.729	0.483	0.553	0.116/0.354
FP	0.197	0.301	0.258	0.852/0.606
TN	0.849	0.753	0.762	0.909/0.887
FN	0.081	0.066	0.069	0.056/0.0069

$$MCC = \frac{(TP * TN - FP * FN)}{\sqrt{(TP + FN) * (TP + FP) * (TN + FP) * (TN + FN)}}$$

From the table above, it indicated PRIME-3D2D is good at prediction of TP (true positives) and TN (true negatives) compared to FP and FN. Instead of what the reviewer said "PRIME-3D2D is good at prediction of TP (true positives) but not TN (true negatives)".

Q: (4) Page 5 line 107, "The accuracy of 3D2D-PRIME is reduced when the template and target are de-redundant by 60% sequence similarity, indicating that some proteins may bind to similar RNAs".

It's not surprising when sequence similarity cutoff affects accuracy of predictions because more high confidence templates may be excluded. Not sure the explanation of "some proteins may bind to similar RNAs" is very informative here.

A: We appreciate this important point. Referring to the reviewer's suggestion, we removed ", indicating that some proteins may bind to similar RNAs.", and it was modified in lines 251-253 with "The accuracy of PRIME-3D2D is reduced when the template and target are de-redundant by 60% sequence similarity."

Q: (5) Minor typos, etc and suggested corrections

Page 3 line 65, "is depended on" -> "depends on"

Page 4 line 83, "expect ACC" -> "except"

Figure 1 legend, first sentence. "..and the result of PRIME-3D3D". Is it meant for "PRIME-3D2D"?

A: Thank you for your suggestions. According to your suggestion, we revised the grammar questions you suggested and made carefully grammatical changes to the entire article. This is reflected in:

- 1) In lines 167 and 385, we changed "is depended on" to "depends on";
- 2) In line 216, we changed "expect ACC" to "except ACC";
- 3) "..and the result of PRIME-3D3D". It is meant for "PRIME-3D2D", and we changed "PRIME-3D3D" to "PRIME-3D2D" in line 604.

Reviewer #2 (Remarks to the Author):

The authors present a novel approach, PRIME-3D2D, to predict protein-RNA interaction considering protein 3D structure and RNA 2D structure. This method was applied to the binding sites at a transcriptome wide level. It would provide a possible way to study protein-RNA interactions at a large scale. However, some concerns need to be addressed.

Major Points:

Q: 1. The paper does not read like a Correspondence. It should be reformat as a research article and more detailed Results/Methods need to be provided carefully.

A: On the advice of the reviewer, we changed the format of the article to research article and added the details in Results/Methods. Please see the revised article for details.

Q: 2. RNA secondary structure will be affected by different protein binding at different conditions/cells. Is it reasonable to predict RNA-protein binding derived from CLIPdb using structure information derived from SeqFold?

A: We appreciate this important point. We think this treatment is reasonable for the reasons as follow. Seqfold is one of the best methods for predicting RNA secondary structure currently which is based on experimental data. Although the secondary structure of RNA may change after binding to RBP, we use the secondary structure information of RNA in a certain state. This structural information can be derived from PDB, sequencing, or prediction methods. We assume that the secondary structure of RNA before and after binding RBP has not changed. Of course, if the secondary structure of RNA is unknown or the structure changes greatly after binding to RBPs, it may cause our method to be unsuccessful. In Figure 5, it shows an example that PRIME-3D2D can be used for genome-scale binding site prediction. This example predicts the binding site of protein (GBP2) on RNA (YKL068W-A). The target protein GBP2 uses Npl3 as the template and the target RNA YKL068W-A employs YNL113W as the template. The predicted TP, FP, TN and FN are 0.952, 0, 0.958 and 0.002 respectively.

Q: 3. PRIME-3D2D integrates RNA secondary structure and RBP binding site data of yeast genome. However, a large amount of RNA secondary structure and RBP binding site data of other species have been curated and available in public databases. The authors need to apply this method to more species and more data. For instance, the RBP binding sites were collected from a subset of CLIPdb. More updated CLIPdb/POSTAR2 datasets need to be

included. And more details about how to integrate RBP binding sites need to be described.

A: The reviewer brings up a good point. There is not enough time to calculate the result of other species due to time issues. But we have plans to apply the PRIME-3D2D method to other species included in the updated database POSTAR2, which will be demonstrated in the next work.

The details about how to integrate RBP binding sites were in the "Method" section of the main text. To put it simply, we think that the BED file in CLIPdb/POSTAR2 contains the native binding sites of RBPs on RNA. Therefore, we selected the RNAs that obtained the secondary structure by experimental methods and simultaneously bind to RBPs in CLIPdb/POSTAR2 to predict binding sites. Then, the predicted binding site can be compared with the native binding site to assess the accuracy of the prediction.

Q: 4. The authors mentioned that PRIME-3D2D was derived from the protein-RNA 3D3D-model. I understand that Protein-RNA 3D3D-model can't be used on genome wide for the lack of RNA 3D structures. However, for those data with 3D structures, the authors should compare and show the performances of both PRIME-3D2D and protein-RNA 3D3D.

A: Thanks for the suggestion of reviewer, we made the following comparison between PRIME1.0, PRIME2.0 and PRIME-3D2D. 1, On NRBC439 data set, we compared them in complex structure modeling; 2, On the data sets of RB75/RB172/RB344, we compared them in the performance of binding sites prediction. The results are as follows:

Figure 3. The result of PRIME1.0, PRIME2.0 and PRIME-3D2D on modeling complexes on NRBC439 dataset. The result showed that PRIME-3D2D/PRIME/PRIME2.0 get relatively close accuracy when constructing three-dimensional complex structures. The subtle difference may be that the RNA comparison procedures are different. PRIME uses SARA, PRIME2.0 uses RAlign, and PRIME-3D2D uses LocARNA. We put it in the lines 189-191 and the figure 3 of the revised version.

The binding sites of PRIME2.0 were calculated from the predicted complex structure model. The interface residue was defined by $< 4.5 \text{ \AA}$ distance between any heavy atom of the protein and any heavy atom of the RNA. And the binding sites of PRIME-3D2D were predicted based on the alignment of target and template (Figure 2A in the revised version). Table1 and Table2 are the comparison results of PRIME2.0 and PRIME-3D2D on binding site on RNA and binding site on protein prediction, respectively.

Table 1. Performance of the PRIME2.0/PRIME-3D2D for protein-binding site prediction

Method	Data Set	Performance (binding site on RNA)						
		ACC	SN	SP	MCC	Strength	F-Measure	PRE
PRIME-3D2D	RB75	0.849	0.965	0.778	0.721	0.872	0.828	0.725
	RB172	0.694	0.769	0.651	0.404	0.710	0.647	0.559

	RB344	0.762	0.814	0.725	0.532	0.770	0.742	0.682
	RB75	0.940	0.531	0.972	0.532	0.752	0.563	0.599
PRIME2.0	RB172	0.781	0.546	0.889	0.467	0.717	0.610	0.692
	RB344	0.840	0.599	0.903	0.509	0.751	0.610	0.621

From table 1, we can see that on the RB75 and RB344 datasets, PRIME-3D2D is superior to PRIME2.0 in protein-binding sites prediction. It illustrated that the secondary structure of RNA is helpful to find more accurate binding sites in protein-binding site prediction.

Table 2. Performance of the PRIME2.0/PRIME-3D2D for RNA-binding site prediction

Method	Data Set	Performance (binding site on protein)						
		ACC	SN	SP	MCC	Strength	F-Measure	PRE
	RB75	0.850	0.90	0.811	0.706	0.856	0.840	0.787
PRIME-3D2D	RB172	0.772	0.884	0.714	0.567	0.799	0.726	0.616
	RB344	0.801	0.889	0.747	0.617	0.818	0.772	0.682
	RB75	0.918	0.876	0.938	0.812	0.907	0.872	0.869
PRIME2.0	RB172	0.900	0.687	0.946	0.650	0.817	0.710	0.734
	RB344	0.911	0.806	0.940	0.742	0.873	0.799	0.792

From table 2, we can see that PRIME2.0 is superior to PRIME-3D2D in RNA-binding sites prediction. It illustrated that the three-dimensional structure of RNA is better than that of the secondary structure to find more accurate binding sites on protein.

We put the table 1 and table 2 on the revised version and the introduction is placed on lines 220-235 of the revised version.

Q: 5. The authors mentioned two approaches detecting the ability of template searching, LocARNA and RNA2dA. The authors gave a detailed introduction to RNA2dA, but not to LocARNA.

A: RNA2dA is one of the work presented in this work and LocARNA was developed by Will *et al.* (Will, et al., 2007), so there were relatively more descriptions of RNA2dA. Considering the reviewer's suggestion, we added descriptions of LocARNA in the revised version. The details are as follows: LocARNA combines RNA sequence and RNA secondary structure information to align RNAs in globally or locally. It is currently one of the most effective and convenient methods for RNA secondary structure alignment. This method can classify RNAs according to RNA secondary structure. Hence,

this method can be used to align the secondary structure of the target RNA to the known secondary structure of RNA, to find a known RNA similar to the target. It was added in lines 355-360 in the revised version.

Q: 6. Interaction RMSD (iRMSD) and 3D2D-score are two important index values of PRIME-3D2D. The similarity of binding mode between the target and the template is measured by interaction RMSD (iRMSD) and 3D2D-score. In Supplementary Fig. 5, the authors show that iRMSD is plotted against 3D2D-score. Are these two indicators the same? What is the relationship of them?

A: We appreciate this important point. iRMSD and 3D2D-score are different. iRMSD describes the geometric similarity between two complex structures. However, 3D2D-score describes the similarity between monomers and does not require a complex structure.

Q: 7. Figure 1.B, C and D show the results of PRIME-3D2D (top 1) comparing with the state-of-the-art methods for predicting RNA-binding site in protein on three PDB datasets (RB75, RB172 and RB344, respectively). The state-of-the-art methods for predicting RNA-binding site in protein, like PiRaNhA, BindN and so on, should be described in the Methods. And more details need to be provided in the results. For instance, why are the control algorithms applied differently in RB75, RB172 and RB344? Why do the control algorithms have different degrees of vacancy in SN, SP, PRE, ACC, Strength, F-measure and AUC? It is known from the Figure 1.B, C and D that MCC is the evaluation index that can be calculated by all algorithms. Why not use it as the optimal evaluation index?

A: Thanks for the reviewer's suggestions. We checked PiRaNhA, BindN and other methods. However, these methods are not available at present. Hence we used the results from the review literature(Si, et al., 2015) , in which the authors evaluated the softwares for predicting RNA binding sites on proteins on the RB75/RB172/RB344 data sets. We changed the article to the research format, so we put the introduction of other binding site prediction methods in the "Introduction" section, specifically on lines 57-81 in the revised article. The details of the results on the RB75/RB172/RB344 data sets are in lines 209-220. For a clearer explanation, we copy the content from the text. "On PDB scale, PRIME-3D2D(PDB) was compared with the RNA-binding site predictors on independent testing sets (RB75, RB172 and RB344). These three independent testing sets and the results of other RNA-binding site predictors were grabbed from the review article(Si, et al., 2015), in which the authors benchmarked the softwares for predicting RNA binding sites on proteins. From Figure 2B, we can see that in the RB75 dataset, the Meta-predictor has the highest AUC, and PRIME-3D2D achieves the best value among all the other evaluation indexes. In the dataset

RB172(Figure 2C), all evaluation indexes of PRIME-3D2D are better than other methods except ACC. In the RB344 dataset (Figure 2D), PRIME-3D2D reaches the lowest SP, but obtains the highest values in the remaining evaluation indexes. Taken together, PRIME-3D2D achieves the best results on these three datasets by using MCC as the main indicator and others as auxiliary evaluation indexes.”.

When comparing PRIME-3D2D with other methods, we used MCC as the main indicator and others are used as auxiliary evaluation indexes. We also concluded that PRIME-3D2D is superior to other methods because of that PRIME-3D2D's MCC is higher than that of other methods. We changed our statement in lines 218-220, from “Taken together, PRIME-3D2D achieves the best results on these three datasets.” to “Taken together, PRIME-3D2D achieves the best results on these three datasets by using MCC as the main indicator and others as auxiliary evaluation indexes.”

Minor Points:

We found some typos in this manuscript. Some examples are listed here:

Q: 8. In page 5 of manuscript, “In summary” does not need to be bold.

A: Thank you for your suggestions. According to your suggestion, we revised the grammar questions you suggested and made carefully grammatical changes to the entire article. On line 247 of the revised version, we changed the bold "In summary" to the non-bold "In summary".

Q: 9. In page 1 of supplementary materials and methods, “interactionome” should be “interactome”.

A: Thank you for your suggestions. On lines 8,258 and 261 of revised version, we changed "interactionome" to "interactome".

Q: 10. In page 8 of supplementary materials and methods, “Figule” should be “Figure”.

A: Thank you for your suggestions. We have changed the “Figule” to “Figure”.

Reviewer #3 (Remarks to the Author):

Q:1. In RNA2dA, the author used the matrix PAM, not BLOUSM, which is different from the Beagle method. Please explain the difference between them and why choose PAM.

A: The reviewer brings up a good point. The Beagle used the PAM-like (point accepted mutation) matrix to score the alignment, which is derived from the

BEAR representation of RNA secondary structure (*Nucleic Acids Res.* 2014 Jun;42(10):6146-57). But in RNA2dA, we used BLOSUM-like matrix instead of PAM-like matrix because we cannot get the PAM matrix used in the Beagle program.

The main differences between PAM and BLOSUM scoring matrix are: 1. PAM matrix was based on evolutionary theory; BLOSUM was not based on an explicit evolutionary mode, but the proteins/nucleic acids corresponding to BLOSUM matrix have a common ancestor(Mount, 2008); 2. The PAM model was designed to track the evolutionary origins of proteins, whereas the BLOSUM model was designed to find their conserved domains(Mount, 2008); 3. PAM matrices were based on global alignments of closely related proteins but BLOSUM matrices were based on local alignments(Choudhuri, 2014).

Q: 2. The author have developed a tool of RNA-binding site prediction, named PRIME, which needs the information of protein 3D structure and RNA 3D structure. The novel method named PRIME-3D2D (protein 3D structure and RNA 2D structure needed) seemed less input information? The reader wonder to know if there are any difference or advance from PRIME? Please explain it in detail.

A: We appreciate this comment. The differences between PRIME and PRIME-3D2D are mainly in: 1. PRIME requires the 3D structure of the RNA as input, while PRIME-3D2D requires the 2D structure of the RNA; 2. PRIME is used for prediction complex structure. When applied to the PDB scale, the PRIME-3D2D model can predict both the complex structure and the binding site; on the genome scale, PRIME-3D2D is used to predict the binding site of proteins on RNA; 3. When searching templates in RNA, the algorithm is different. SARA/RMalign is used for RNA alignment in PRIME1.0/PRIME2.0, and LocARNA is used for structural alignment in PRIME-3D2D; 4. The finally complex structure scores are different. PRIME 1.0/2.0 uses the lower score between protein and RNA as the complex structure score, but PRIME-3D2D employs a linear combination between protein and RNA structure scores. And it was present in lines 314-325 of the revised version.

Q: 3. NRBC439 is split into two parts. 80% with an older deposit date are designated as the templates (NRBC349), and 20% with a newer deposit date are designated as targets (NRBC90). The rule of deciding which data is template and which data is target is important. Please tell the rule. If necessary, cross-validation should be used here.

A: Thanks for your comment. We divide the targets and templates according to the time of the complex structures deposited in the PDB. Our purpose is to predict the new complex structure based on the known complex structures, so it is reasonable to use such a division method. In addition, it is also to be consistent with the division method in the PRIME(Zheng, et al., 2016)/PRIME2.0(Zheng, et al., 2019). And it was present in lines 339-344 of

the revised version.

Q: 4. I submitted a task in the website of PRIME-3D2D, but unfortunately I did not receive the result. Please check whether the website works.

A: Thanks for your advice. Around the Spring Festival, the server was disconnected for a few days. Now it has been reconnected, you can re-deliver the task for calculation. We submitted the task to the server, and the results showed that the server was working normally. For the PDB scale, you can input the structure of the protein and RNA, or you can input the PDB ID of the protein and RNA; for the genome scale, for the protein, you can input the structure of the protein or the PDB ID of the protein, and for RNA, you need to input RNA sequence information and RNA secondary structure information, you can refer to the example provided on our website. Due to the large number of templates, the calculation time will be relatively long. If you provide an email address, you will receive a reminder email after the task is completed.

Q: 5. NRBC439, as the main dataset in this work, I think it is necessary to divide the data into train data and test data using 10-fold cross-validation, in order to ensure the reliability of the prediction results.

A: Thanks for your comment. PRIME-3D2D is to predict new complex structures or binding sites based on known structures and we performed an all-to-all comparison on the NRBC439 data set, so we think that ten-fold cross-validation is not required.

References

Choudhuri, S. Sequence Alignment and Similarity Searching in Genomic Databases. In, *Bioinformatics for Beginners*. 2014. p. 133-155.

Mount, D.W. Comparison of the PAM and BLOSUM Amino Acid Substitution Matrices. *CSH Protoc* 2008;2008:pdb ip59.

Si, J., *et al.* Computational Prediction of RNA-Binding Proteins and Binding Sites. *Int J Mol Sci* 2015;16(11):26303-26317.

Will, S., *et al.* Inferring noncoding RNA families and classes by means of genome-scale structure-based clustering. *PLoS Comput Biol* 2007;3(4):e65.

Zheng, J., *et al.* Template-Based Modeling of Protein-RNA Interactions. *PLoS Comput Biol* 2016;12(9):e1005120.

Zheng, J., *et al.* RMAalign: an RNA structural alignment tool based on a novel scoring function RMscore. *BMC Genomics* 2019;20(1):276.

REVIEWERS' COMMENTS:

Reviewer #1 (Remarks to the Author):

The authors properly addressed my concern by providing additional data. In particular, a comparison of performance between PRIME-3D2D and previous version PRIME2.0 is provided in Tables 1 & 2, as well as Figure 3.

Reviewer #2 (Remarks to the Author):

The authors have addressed most of my concerns.

1. The author refined the article but the overall logic of the article is still not clear enough.
2. The author assumed that the secondary structure of RNA before and after binding RBP has not changed. It at least needs to be discussed in the discussion section

REVIEWERS' COMMENTS:

Reviewer #1 (Remarks to the Author):

Q: The authors properly addressed my concern by providing additional data. In particular, a comparison of performance between PRIME-3D2D and previous version PRIME2.0 is provided in Tables 1 & 2, as well as Figure 3.

A : Thank you for evaluating our revised version

Reviewer #2 (Remarks to the Author):

Q: The authors have addressed most of my concerns.

A: Thank you for your overall evaluation of our revised version.

Q: 1. The author refined the article but the overall logic of the article is still not clear enough.

A: Thanks for your comment. We conducted a comprehensive review of the article.

Q: 2. The author assumed that the secondary structure of RNA before and after binding RBP has not changed. It at least needs to be discussed in the discussion section.

A: We appreciate the reviewer to figure out this important point. Based on the reviewer's suggestion, we added the following in the method section and discussion section:

Method part: "We assume that the secondary structural state of RNA remains the same before and after binding to the protein." (in the lines 348-349 of the revised version).

Discussion section : "In this process, we use the secondary structure of the RNA predicted by SeqFold, which is based on experimental data, and assume that the secondary structure of RNA before and after binding RBP has not been changed to simplify the model." (in the lines 262-265 of the revised version).